# Highlighting In Vitro the Role of Brain-like Endothelial Cells on the Maturation and Metabolism of Brain Pericytes by SWATH Proteomics

**DOI:** 10.3390/cells12071010

**Published:** 2023-03-25

**Authors:** Camille Menaceur, Johan Hachani, Shiraz Dib, Sophie Duban-Deweer, Yannis Karamanos, Fumitaka Shimizu, Takashi Kanda, Fabien Gosselet, Laurence Fenart, Julien Saint-Pol

**Affiliations:** 1Univ. Artois, UR 2465, Blood-Brain Barrier Laboratory (LBHE), F-62300 Lens, France; 2Department of Neurology and Clinical Neuroscience, Graduate School of Medicine, Yamaguchi University, Ube 755-8505, Japan

**Keywords:** brain pericytes, blood-brain barrier, human syngeneic in vitro model, label-free quantitative proteomics, SWATH, cell-cell communications, cell maturation

## Abstract

Within the neurovascular unit, brain pericytes (BPs) are of major importance for the induction and maintenance of the properties of the blood-brain barrier (BBB) carried by the brain microvessel endothelial cells (ECs). Throughout barriergenesis, ECs take advantage of soluble elements or contact with BPs to maintain BBB integrity and the regulation of their cellular homeostasis. However, very few studies have focused on the role of ECs in the maturation of BPs. The aim of this study is to shed light on the proteome of BPs solocultured (hBP-solo) or cocultured with ECs (hBP-coc) to model the human BBB in a non-contact manner. We first generated protein libraries for each condition and identified 2233 proteins in hBP-solo versus 2492 in hBP-coc and 2035 common proteins. We performed a quantification of the enriched proteins in each condition by sequential window acquisition of all theoretical mass spectra (SWATH) analysis. We found 51 proteins enriched in hBP-solo related to cell proliferation, contractility, adhesion and extracellular matrix element production, a protein pattern related to an immature cell. In contrast, 90 proteins are enriched in hBP-coc associated with a reduction in contractile activities as observed in vivo in ‘mature’ BPs, and a significant gain in different metabolic functions, particularly related to mitochondrial activities and sterol metabolism. This study highlights that BPs take advantage of ECs during barriergenesis to make a metabolic switch in favor of BBB homeostasis in vitro.

## 1. Introduction

Although a discrete component of a brain microvascular multicellular entity named the neurovascular unit (NVU), the brain pericytes (BPs) have a major role in the induction and maintenance of the blood-brain barrier (BBB) properties [1,2,3]. BPs are recruited at early stages of brain development by brain endothelial cells (ECs) secreting platelet-derived growth factor-β (PBGF-β) [4]. This recruitment promotes pro-angiogenic processes, the formation of blood vessels and the development of a common basement membrane, which occurs when ECs and BPs are close or in contact through the secretion of transforming growth factor-β (TGF-β) by BPs [5,6,7,8]. Then, brain microvessel ECs mature in response to the secretion of angiopoietin-1 (Ang1) by BPs [1,2,9], the activation of Notch1-4/Smad-4 signaling pathway [10,11,12] and also some possible contact sites referred to as peg-and-socket junctions facilitating the exchanges of soluble factors [13,14,15,16]. Thus, gradually, ECs acquire the mature and functional BBB main features some days after birth [3], characterized by (i) junctional complexes composed by apical tight junctions, median and basolateral adherent junctions and/or Ca^2+^-mediated junctions, (ii) limited aspecific transports, transcytosis routes being restricted to specific transporters, receptor-mediated clathrin- or caveolin-dependent forms or adsorptive events, (iii) a limited passive diffusion of compounds due to efflux pumps and metabolic enzymes [3,17,18,19]. All along life, BP’s physiology and fate are directly linked to BBB maintenance and leakage in pathological contexts. Indeed, the loss of BPs or their alteration observed in neuropathological conditions such as Alzheimer’s disease leads to a decrease of stability and integrity of the BBB [20]. Moreover, due to the expression of contractile proteins [21], BPs regulate regional blood flow in brain microvessels and capillaries, but this regulation remains modest compared to smooth muscle cells in upstream arteriolar networks [22,23,24,25]. However, the abnormal contractility of BPs has consequences for the proliferation and pro-angiogenic capacity of ECs. Indeed, it has been reported that mutation of the myosin phosphatase-RhoA interacting protein (MRIP) results in increased contractile activity and disorganisation of the pericyte cytoskeleton into stress fibers, leading to deregulation of the ECs cell cycle and pro-angiogenic behavior [26]. BPs exhibit, therefore, a minute contractility, and an abnormal contractile capacity is consistent with a destabilization of local cerebral blood flow and the BBB properties [27].

In a so-called traditional manner, studies on BPs only focused on their role in the induction and maintenance of BBB properties and homeostasis of ECs in physiological and pathological conditions [28,29]. However, very few studies have focused on the contribution of ECs to BPs and their development or maturation. Thus, the purpose of this study is to investigate in vitro the proteome of human BPs cultured alone (hBP-solo condition) or cocultured with ECs (hBP-coc) for the time required to induce the BBB main features on ECs, i.e., 6 days for the human in vitro syngeneic BBB model used [29,30]. For this, we first established the protein libraries of hBP-solo and hBP-coc, and then we quantified by a sequential window acquisition of all theoretical mass spectra (SWATH) analysis the proteins enriched in each condition to determine the possible maturation and functional switches in BPs after the induction of the BBB phenotype on ECs in vitro.

## 2. Materials and Methods

### 2.1. Cells and Cell Culture

The human brain pericytes (hBPs) were obtained from the Pr. Takashi Kanda’s team (Department of Neurology and Clinical Neuroscience Clinical Neuroscience, Yamaguchi University Graduate School of Medicine of Medicine, Yamaguchi University, Ube, Japan). They were isolated from a patient who suddenly died from a heart attack [31]. The study protocol for human tissue was approved by the ethics committee of the Medical Faculty (IRB#: H18-033-6), University of Yamaguchi Graduate School and conducted in accordance with the Declaration of Helsinki, as amended in Somerset West in 1996. Written informed consent was obtained from the family of the participant before entering the study.

After isolation from brain tissue, pericytes were immortalized by transfection of retrovirus vectors encoding the temperature-sensitive SV40 T antigen (tsA58) and encoding human telomerase (hTERT) [31]. The hBP cell line was previously characterized for so-called pericyte markers such as Desmin, PDGFR-ß and α-SMA [32].

After thawing, 6.25 × 10^5^ hBPs used between passage 15 and 25 are seeded in gelatin-coated (Sigma-Aldrich-Aldrich, Saint-Louis, MS, USA, G-2500) 100-mm-diameter Petri dishes and cultured in DMEM (Dulbecco’s modified Eagle’s medium; Life Technologies) supplemented with 4.5 g/L glucose, 10% fetal calf serum (FCS, Life Technologies, Waltham, MA, USA), 2 mM L-glutamine, and 1% antibiotics (penicillin and streptomycin).

Human hematopoietic stem cells were obtained from umbilical cord blood as previously described [29,30]. Informed consent was obtained for the collection of human umbilical cord blood. The protocol was approved by the French Ministry of Higher Education and Research (CODECOH Number DC2011-1321), and all experiments were conducted in accordance with the approved protocol. Briefly, blood was centrifuged in a Ficoll gradient to isolate the mononuclear cells. CD34^+^ hematopoietic stem cells were then sorted on MACS columns [29,33] and cultured in gelatin-coated 24-well plates in EGM-2 medium (Lonza, Basel, Switzerland) supplemented with 20% FCS and 50 ng/mL VEGF (PeproTech Inc, Cranbury, NJ, USA). After 15 to 20 days, CD34^+^ cells were differentiated into endothelial cells (ECs) and cultured in 100-mm diameter gelatin matrix-coated Petri dishes (0.2%) in ECM-5 medium corresponding to Endothelial Cell Medium (ECM, ScienCell, Carlsbad, CA, USA) supplemented with 5% FCS, 2 mM L-glutamine, 50 µg/mL gentamycin and 1% Endothelial Cell Growth Supplement (ECGS, ScienCell, Carlsbad, CA, USA) [29,30]. Cells were cultured at 37 °C in a humid atmosphere with 5% CO_2_.

On the 6th day of coculture, hBP were treated with trypsin/EDTA and seeded at 6.25 × 10^5^ cells in a Petri Dish (Costar, Transwell 3419, Corning, NY, USA), or 8 × 10^4^ cells on a coverslip of 18 mm (ThermoFisher Scientific, Waltham, MA, USA, 0111580) in a 12 well plate (Costar, 3512, Corning, NY, USA). At confluence, CD34^+^ endothelial cells are also treated with trypsin/EDTA and seeded on filtered inserts (0.4 µm, 75 mm, Costar 3419) previously covered with Matrigel^TM^ (Corning, NY, USA) at a rate of 3.1 × 10^6^ cells/filter. Filters with endothelial cells were placed on top of wells containing hBPs. The cells are grown in coculture for 6 days. 

### 2.2. Sample Preparation for Label-Free Mass Spectrometry (MS)

Protein extracts from hBPs were prepared from two independent batches after 6 days of culture alone (hBP-solo) or cocultured with CD34^+^ endothelial cells (hBP-coc). hBP-solo and hBP-coc were treated with Accutase (StemCell Tech., Vancouver, BC, Canada, #07920) and the detached cells recovered in pellet by centrifugation at 300× *g* for 5 min at 4 °C. The cells were then rinsed three times with phosphate buffer saline (PBS, 8 g/L NaCl, 0.2 g/L KCl, 0.2 g/L KH_2_PO_4_, 2.87 g/L NaHPO_4_ (12 H_2_O), 0.1 g/L CaCl_2_, 0.1 g/L MgCl_2_ (6H_2_O), pH: 7.4) and centrifuged for 5 min at 300× *g* at 4 °C, and the final pellets were stored at −80 °C until preparation for MS analysis. hBP total proteins were extracted with a 50 mM Tris-HCl buffer (pH 7.5) containing 6M guanidine-hydrochloride for 5 min at 4 °C (ice). Samples were purified from cell debris and contaminating DNA with Total RNA Protein Isolation kit (Sigma-Aldrich, Germany) following manufacturer’s recommendations. The protein concentration was determined using the Quick Start Bradford dye reagent (Biorad, Hercules, USA). A 100 µg sample of protein was reduced with 25 mM dithiothreitol (DTT, Sigma-Aldrich, *w*/*v*) in 25 mM ammonium bicarbonate (NH_4_HCO_3_) for 20 min at 56 °C, alkylated in 50 mM iodoacetamide (*w*/*v*) in 25 mM NH_4_HCO_3_ for 20 min in the dark at room temperature and purified by ice cold 80% acetone precipitation, overnight at −20 °C. The precipitate was recovered by centrifugation at 14.000× *g* for 10 min at 4 °C and digested in 2 µg Trypsin (Promega, Madison, WI, USA) at 37 °C overnight in 50 mM NH_4_HCO_3_ (*w*/*v*) pH 8.5 with an enzyme/substrate ratio of 1/50. The enzymatic reaction was stopped in 0.2% (*v*/*v*) formic acid in water. Peptides were desalted, trapped, and concentrated using the HyperSep SpinTip Microscale C18 (ThermoFisher Scientific, Waltham, MA, USA). Samples were dried and resuspended in a solution of 2% acetonitrile (ACN) 0.1% (*v*/*v*) formic acid in water. Peptide concentration was measured by a Quantitative Colorimetric Peptide Assay (Thermo Fisher Scientific, Waltham, MA, USA), and samples prepared with a final concentration of 1 µg/µL before MS analysis.

### 2.3. Generation of hBP Protein Spectral Library by Data-Dependent Analysis (DDA) MS

To generate the protein libraries, 2 µg samples of protein from each condition were analyzed using an LC–MS/MS system composed by an AB SCIEX TripleTOF 5600+ mass spectrometer (AB Sciex, Foster City, CA, USA) and an Ekspert nanoLC 400 System. Each sample was injected 5 times (N = 2, *n* = 10) and a PepCalMix LC-MS solution (AB Sciex) was used for recalibration after each injection allowing for maintenance of a mean mass error below 10 ppm.

After each injection and for chromatographic separation in micro-flow mode, samples were loaded in a Trap column (Luna Omega 5 µm Polar C18 100 Å, Micro Trap 20 × 0.5 mm, Phenomenex, Torrance, CA, USA) with a rate of 10 µL/min for 3 min and separated by an Eksigent Chrom XP-C18 HPLC reverse phase column (0.3 × 150 mm, 120 Å, particle size 3 μm) using a variable gradient of the mobile phase composed of 0.1% (*v*/*v*) formic acid in 2% ACN 98% water (phase A) and 0.1% (*v*/*v*) formic acid in 98% ACN 2% water (phase B) with a flow rate of 5 μL/min. The following gradient details are given for the percentages of phase B in phase A only: from 3 to 25% for 68 min, from 25 to 35% for 5 min, 35 to 80% for 2 min, 80% for 3 min, 80 to 3% for 1 min, 3% for 6 min. The temperature of the autosampler and column were maintained at 8 °C and 35 °C, respectively. The column eluent was directed into the AB Sciex TripleTOF 5600+ system, then was operated in positive ion mode by electrospray ionization using the DuoSpray^®^ and Turbo V^®^ ionization sources and controlled by the Analyst software (version 1.7.1). Ionization parameters were as follows: ISVF = 5500, GS1 = 15, GS2 = 15, CUR = 25, TEM = 150. MS spectra were acquired in high sensitivity (HS) mode for 250ms from 400 to 1250 m/z and MS/MS scan from 100 to 1500 m/z (60 ms accumulation time, 20 ppm mass tolerance, rolling collision energy). The generation of protein libraries was performed using the ProteinPilot software (AB Sciex, version 5.0.2) with a UniProt human-filtered proteome database (February 2021), specifying iodoacetamide and methionine oxidation as variable modifications. The detected protein threshold was set to 10% (unused ProtScore > 0.05). Identification data for the two independent biological samples were merged to generate the so-called hBP-solo and hBP-coc protein libraries.

### 2.4. Data-Independent Analysis (DIA)/Sequential Window Acquisition of All Theoretical Mass Spectra (SWATH)

SWATH-MS was done using the same material and instrumental setup as described for the DDA-MS analysis with some modifications. Briefly, a 100-variable-window setup was generated using the SWATH^®^ Variable Window Calculator 1.1 (AB Sciex) with a 1 m/z window overlap on the lower side of the window. The MS1 survey scan was acquired from 400–1250 m/z for 50 ms and MS2 spectra were acquired in high-sensitivity mode from 400–1500 m/z for 30 ms. The total cycle time was ~3.1 s. The collision energy used in SWATH-MS was that applied to a doubly charged precursor centered in the middle of the isolation window calculated with the same collision energy equation for DDA, and with a CES of 10 eV. For the analyses conducted in the capillary flow and microflow rate, the SWATH-MS data were recorded as described for the DDA-MS.

The SWATH-MS data analysis was performed using PeakView software (version 2.2, AB Sciex) for a local SWATH-MS processing workflow. The endogenous hBP peptides were extracted according to the precursor m/z, intensity and confidence of identification across the entire time range, and the best scoring peak groups were used for retention time (RT) calibration. The spectral library and SWATH-MS data were loaded into the SWATH™ Processing microApp. The peak groups were extracted with a 99% peptide confidence threshold and a 1% peptide FDR threshold. The XIC extraction window and fragment ion mass tolerance were set to 10 min and 50 ppm, respectively. XIC calculation has been done for the top 5 peptides of all proteins identified. After data extraction, the results were imported into MarkerView™ (version 1.2.1.1) for further data processing and normalization, and analyzed according to the total area sums. The area under the XIC curves of peptides were individually normalized based on a summed area of all peptides for each sample. Principal component analysis (Pareto mode for scaling) was performed on samples based on expression to visualize sample clustering, and group comparisons were performed with a Student’s t-test to point out fold change (FC) and *p* values. Data were filtered for *p* value < 0.01 and FC > 2 (log(FoldChange) > 0.3) or FC < −2 (log(FoldChange) < 0.3). All the quantification results for proteins with less than 2 peptides were automatically excluded, and the accession number of the proteins that met both conditions were analyzed using MetaScape [34] and the STRING database [35]. The interaction maps from the STRING database were obtained for an accuracy of 0.4 and customized by Cytoscape (version 3.9.1) [36,37] to point out the proteins of interest.

### 2.5. Dosage of Total Cell Cholesterol

After 6 days, hBP-solo and hBP-coc were treated or not with acetylated LDL (ac-LDL, 25 µg/mL, Invitrogen, Waltham, MA, USA, L35354) during 3h and recovered in dry pellet after accutase detachment to be prepared for cholesterol assay. The dry pellet is rinsed twice in Ringer-Hepes buffer (RH, 150 mM NaCl, 5.2 mM KCl, 2.2 mM CaCl_2_, 0.2 mM MgCl_2_-6H_2_O, 6 mM NaHCO_3_, 5 mM HEPES, 2.8 mM glucose, pH: 7.4). The cells were then lysed in lysis buffer and centrifuged for 10 min at 13,000× *g* at room temperature. The supernatants were kept and heated for 5 min at 50 degrees. The samples were then dried in the vacuum for 30 min. The samples were then assayed using a Cholesterol Quantitative Kit (Sigma-Aldrich, MAK043). Briefly, this assay uses a coupled enzyme giving a colorimetric (570 nm)/fluorometric (λex = 535 nm/λem = 587 nm) product, proportional to total cell cholesterol and more precisely free cholesterol and cholesterys esters. The measurement of the total cell cholesterol was normalized by µg of protein in each hBP sample.

### 2.6. Statistical Analysis

The results are indicated as the mean ± SEM and analyzed by Student’s *t*-test or one-way ANOVA test followed by multiple comparisons for different conditions. All the statistical tests were performed using Prism Software (GraphPad Software Inc., San Diego, CA, USA).

## 3. Results

### 3.1. Dosage of Total Cell Setup of Protein Libraries from Solocultured and Cocultured Human Brain Pericytes by Data-Dependent Analysis-Mass Spectrometry (DDA-MS)

Figure 1 summarizes the experimental design of the study divided into two major steps: the setup of protein libraries and label-free quantification of proteins enriched in solo-cultured human brain pericytes (hBP-solo) and pericytes cocultured with CD34^+^ endothelial cells used to model the BBB in vitro (hBP-coc) [29,30]. The induction of the BBB on ECs in the coculture condition was confirmed by permeability assays for Lucifer yellow and immunostaining for the tight junction (TJ) protein Claudin-5 and the TJ-associated protein Zonula Occludens-1 (ZO-1, Appendix A). We first established protein libraries from hBP-solo and hBP-coc (Figure 1 Step 1) and identified 2233 proteins (30386 peptides) in hBP-solo and 2492 (31901 peptides) in hBP-coc. 

The Venn diagram for these identified proteins showed that 2035 proteins are common in hBP-solo and hBP-coc conditions, 198 were specifically identified in hBP-solo and 457 in hBP-coc (Figure 2A, Appendix A).

A comparative gene ontology (GO) analysis was conducted based on the enriched proteins in both conditions, and showed on one hand that proteins enriched in hBP-solo are preferentially involved in developmental, growth, and locomotion processes compared to hBP-coc (Figure 2B). These functions linked to actin filament-based processes and positive regulators of cell motility (Figure 2C). On the other hand, hBP-coc enriched proteins are associated with biological and cellular processes (Figure 2B), such as vesicle-mediated transport (Figure 2C). Moreover, proteins enriched in hBP-coc are highly linked to localization GO entry compared to hBP-solo. Altogether, these data suggest a trend for cell differentiation or speciation in hBP-coc with improved specialized cellular and metabolic functions and depressed immature functions such as cell growth, development, and locomotion.

### 3.2. Sequential Window Acquisition of All Theoretical Mass Spectra (SWATH) Data Independent Analysis-Mass Spectrometry for hBP-Solo and hBP-Coc Proteins

To rigorously quantify the proteins enriched in hBP-solo and hBP-coc cell total lysates, we opted for a sequential window acquisition of all theoretical mass spectra (SWATH) approach, a label-free method to quantify the proteins with high accuracy and reproducibility [38]. We led two independent SWATH analysis for two biological batches and crossed the quantification data to sort a list of significant proteins enriched in hBP-solo and hBP-coc. Briefly, each SWATH batch was set up considering as significant candidates the quantified proteins with *p*-values inferior to 0.01 and fold changes superior to 2 for hBP-coc and inferior to 2 for hBP-solo. Figure 3A shows the Volcano plots obtained for each biological batch using these restrictive selection criteria.

We quantified a reproducible number of proteins in the hBP-solo condition with 78 and 85 proteins for Batch 1 and Batch 2, respectively, as well as in the hBP-coc condition (106 and 116 proteins, Appendix A). Then, to promote the strength of this quantitative approach, we combined both batches to sort the proteins quantified in both SWATH approaches with a mean of expression superior to 2 in the hBP-coc condition or inferior to 2 in hBP-solo. Figure 3B compares the number of significant candidates for each condition for each SWATH analysis and after the combination of both. We observed a decrease of significant candidates in the hBP-solo condition by 38%, reducing the number of proteins of interest for this condition to 51 (listed in Appendix A). Similarly, we reduced to 90 the number of significant proteins in the hBP-coc condition (listed in Appendix A). As for the data generated by DDA-MS analysis, we compared the GO from the list of quantitative candidates from the hBP-solo and hBP-coc condition (Figure 3C) and observed that the proteins enriched in the hBP-solo condition are in favor of contractility, blood vessel development, cell-cell adhesion processes, Integrin-1 and vascular endothelial growth factor (VEGF)-mediated pathways. In the hBP-coc condition, the quantitatively enriched proteins were involved in cholesterol and mitochondrial metabolism, confirming the trend toward cell differentiation observed in Figure 2C.

### 3.3. hBP-Solo Enriched Proteins Seem to Play in Favor of a Non-Differentiated and Angiogenic Behavior of hBP According to Transgelin-1-Mediated Processes

The Top 20 proteins upregulated in the hBP-solo condition are listed in Table 1.

GO analysis of these proteins showed their involvement in blood vessel development linked to the VEGF signaling pathway, actin-myosin related contractile movements, and cytoskeleton-related proteins referred to as syndecan interactions and focal adhesion GO functions (Figure 4A). The interaction map of these 51 proteins pointed out two main nodes (Figure 4B). The first main node noticed by orange and yellow circles correspond to secreted extracellular matrix components with collagen fibrils such as collagen alpha-1 (I, III, V) chains (COL1A1, COL3A1 and COL5A1), alpha-2 (I, V) chains (COL1A2 and COL5A2) and proteoglycans known to structure or regulate the expression of extracellular matrix components such as Syndecan-2 (SDC2) [39] and Prolyl-4-hydroxylase subunit alpha-2 (P4HA2). The proteins composing the second main node with blue and light-blue circles referred to cell adhesion proteins (Testin or TES, Vinculin or VCL), actin cytoskeleton organizers (TES), actin/myosin components (Tropomyosin alpha1-chain or TPM1) and regulators (Caldesmon or CALD1, Myosin regulatory light polypeptide 6 and 9 or MYL6 and MYL9). Both nodes were linked to a common regulatory protein, Transgelin-1 or TAGLN (pink circle in Figure 4B), known to act on the previously cited processes through a transforming growth factor-β1 (TGFß1)-mediated pathway [40,41,42]. SWATH quantification showed an increased expression of TAGLN in hBP-solo by 7.03 ± 0.979 compared to hBP-coc (Table 1), as well as a TGF-β1-induced protein, transforming growth factor-beta-induced protein ig-h3 (TGFBI), by 2.13 ± 0.144 (light-pink circle in Figure 4B, Appendix A). These data suggest an improvement in TAGLN/TGF-β-mediated processes in hBP-solo, particularly extracellular matrix production, cell motility, adhesion, and contractility. All these functions are in favor of a so-called immature or undifferentiated pericytes.

### 3.4. hBP-Coc Enriched Proteins Are Mainly Linked to Cholesterol, Fatty Acid and Mitochondrial Metabolisms

As for hBP-solo enriched proteins, we explored the main cell functions driven by the Top 20 proteins quantitatively enriched in the hBP-coc condition (Table 2). Notably, GO analysis exhibited a strong metabolic switch of hBP-coc compared to hBP-solo, particularly lipid metabolism in association with de novo cholesterol biogenesis (−log(P) > 17.5), fatty acid/steroid, and sterol metabolisms (−log(P) > 10.0 and 5.5, respectively (Figure 5A)). Indeed, 6 enzymes involved in the initial steps of cholesterol synthesis and in both Bloch and Kandutsch-Russell pathways (detailed in Appendix A) are upregulated in the hBP-coc condition (green and light-green circles in Figure 5B): squalene synthase (FDFT1, 2.25 ± 0.146); methylsterol monooxygenase 1 (MSMO1, 4.01 ± 1.08); Sterol-4-alpha-carboxylate 3-dehydrogenase, decarboxylating (NSDHL, 2.35 ± 0.245); 3-keto-steroid reductase/17-beta-hydroxysteroid dehydrogenase 7 (HSD17B7, 2.89 ± 0.771); 3-beta-hydroxysteroid-Delta(8),Delta(7)-isomerase (EBP, 2.31 ± 0.062); 7-dehydrocholesterol reductase (DHCR7, 3.15 ± 0.001). According to the interaction map in Figure 5B, these enzymes are functionally linked to proteins involved in fatty acid biosynthesis and β-oxidation (in blue and light-blue circles), in mitochondrial activities (yellow and light-yellow circles), and lipid transporters such as low-density lipoprotein receptor (LDLR, 3.47 ± 0.40) and scavenger receptor class B member 1 or SR-B1 (SCARB1, 3.37 ± 1.632) in magenta circles.

Thus, these SWATH analysis favor a gain in metabolic functions in hBP-coc compared with hBP-solo mostly related to mitochondrial activities, i.e., cellular respiration, β-oxidation of fatty acids, and lipid metabolism, for which cholesterol metabolism seems to be higher in hBP-coc compared to hBP-solo. To validate this hypothesis, we measured after 6 days of culture the total cell cholesterol content of hBP-solo and hBP-coc, and we surprisingly did not observe any significant difference between either condition (Figure 6). However, we showed that the loading of hBP-coc in cholesterol brought by acetylated low density lipoproteins (ac-LDL) is significantly increased by 45% compared to hBP-solo. This observation is in favor of a SR-B1 and/or LDLR-mediated cholesterol uptake from ac-LDL since both proteins are upregulated in hBP-coc (Table 2).

## 4. Discussion

The main purpose of this proteomics study was to highlight the benefit of cell-cell communications between BPs and endothelial cells (ECs) during the time required to model in vitro the human BBB by focusing on BPs. In fact, the focus has traditionally been on ECs, the cell type that carries the BBB phenotype and whose appearance is largely the result of interaction with BPs from the embryonic stages throughout a physiological process called barriergenesis [3]. However, it would be logical that BPs also take advantage of this bidirectional cell-cell communication and adapt their functions to the state of development of the human BBB in vitro and, in a broader point of view, the development of the neurovascular unit in vivo.

### 4.1. Solocultured Brain Pericytes Present a Protein Pattern of ‘Naïve’ Proliferating Cells Conducted by TAGLN/TGF-ß Pathway

One of the primary roles of BPs is to promote angiogenesis during the embryonic stages of the development of the cerebral microvasculature, but also, like the smooth muscle cells surrounding large vessels, i.e., arterioles, arteries, venules and veins, to regulate cerebral blood flow at the level of brain microvessels by contractile mechanisms [43,44]. Proliferation of pericytes is a sign of angiogenic or pathological neoangiogenesis events, such as in tumor development [45]. Our data show that the hBP-solo proteome is enriched in proteins involved in cell motility and contraction such as myosin light chain polypeptide 6 and 9 (MYL6, MYL9), tropomyosin alpha-1 chain (TPM1) and proteins regulating cell motility. Caldesmon participates with Vinculin (VCL) in the regulation of myosin/tropomyosin contractility by creating a function bridge with cadherins and actin cytoskeleton [46,47,48,49,50]. Testin (TES) is a scaffold protein involved in cytoskeleton remodeling during cell spreading and proliferation [51,52]. The fact that these proteins are downregulated in hBP-coc indicates that proliferation and contractility activities in hBPs are under the control of ECs. Differentiated BBB ECs restrict BPs proliferation and density [53] in line with the end of the angiogenesis process and forthcoming appearance and stabilization of the BBB phenotype. The progressive reduction in hBPs contractility argues that the contribution of these cells in the regulation of cerebral blood flow appears to be minimal, compared to smooth muscle cells, upstream of the microvascular and capillary beds [22,23]. In other words, hBP contractile activity seems to decrease progressively during BBB maturation. However, this point remains to be further investigated to better define the mode and mechanisms of this regulation by ECs, and how cerebral blood flow could also impact hBP physiology.

Our results also suggest that the hBP-solo proteome supports the production of ECM components such as collagen fibrils and proteoglycans. The production of ECM is essential for cell stabilization, morphology, proliferation, and motility [54], but also to determine cell fate and cellular functions [55]. ECM is also a structural basis for the formation and function of multicellular complexes in physiological and pathological conditions such as the NVU with the common basal membrane shared by BPs and ECs [56,57,58]. This common ECM is crucial to control cell motility and proliferation of the cells composing the complexes [59]. We observed an enrichment in collagen type I, III and V proteins (COL1A1-2, COL3A1; COL5A1-2) in hBP-solo compared with hBP-coc, collagens fibrils, which are linked to the plasma membrane and stabilized by proteoglycans such as Syndecan-2. Overexpression of Syndecan-2, which is also associated with the increased expression of collagens through integrin pathways [39] and improves cell migration as described in melanoma cells [60]. Prolyl-4-hydroxylase subunit alpha-2 (P4HA2) protein expression is also increased in hBP-solo, an ECM enzyme known to generate and stabilize collagen fibers through the generation of 4-hydroxyproline [61]. Its expression is also increased in cells presenting migrating and proliferating behaviors as observed for cancer cells [62]. hBP-coc also exhibited an enrichment in collagen type IV fibers (COL4A2) linked to the function of a mature basement membrane at the BBB level [57,58,63]. In a previous study conducted on a “contact” human in vitro BBB model, whole cell pericyte transcriptomics and proteomics of secreted proteins, Kurmann and et al. [64] previously observed an opposite trend with an improved expression of ECM components in cocultured BPs. This contact in vitro BBB model is based on the coculture of commercially available Human Brain Vascular Pericytes (HBVP, ScienceCell) and human Cerebral Microvascular Endothelial Cell line (hCMEC/D3 cells [65]) on the two sides of a microporous membrane. These differences observed could be explained by (i) the nature of the cell lines and (ii) mode of coculture used. This contact between BPs and ECs is also in favor of the regulation of ECM production by integrin pathways in both cell types [66]. In our conditions, we explored the non-contact role of ECs on BPs whole-cell proteome and observed the enhanced expression of type IV collagen proteins in hBP-coc, which is in line with an induced production by ECs of so-called mature ECM components by BPs. As also suggested by previous studies [5,7], the promotion of ECM components is linked to a TGF-β pathway when BPs are in contact with ECs, and this process is known to be active during the embryonic period [53] which can be referred to as mid-stages of barriergenesis. In our study, we highlighted that TGF-β-mediated cell proliferation, migration, and the production of major ECM components are promoted in hBP-solo and tend to decrease in hBP-coc. Moreover, we showed that Transgelin-1 (TAGLN), protein relying on the production of ECM components and cell motility and proliferation and known to act as a regulator of the TGF-β pathway [40,41,42], is increased in hBP-solo compared with hBP-coc. Altogether, our data suggest that BP-solo show a behavior equivalent to that observed during the early stages of barriergenesis, leading us to qualify them as ‘naïve’ or ‘immature’ brain pericytes with respect to this BBB regulation context. We can also hypothesize that the observed activation of the TGF-β pathway is related to the early stages of BBB development, and that its progressive extinction is a sign of the acquisition of ‘cellular maturity’ related to the maturation of the BBB phenotype, facts observed for the late and terminal stages of the barriergenesis. These data suggest that there is a need to better understand this pathway and, more specifically, its regulation by TAGLN during embryonic development at the level of brain microvessels.

### 4.2. Non-Contact Cocultured Brain Pericytes Present a Metabolic Switch Linked to a ‘Specific Cell Maturation’ Induced by ECs

According to our DDA-SWATH data, hBP-coc present a proteomic profile related to a decrease in their contractile and/or proliferative activity compared to hBP-solo, but especially with a notable gain in mitochondrial activities and sterol metabolism. In fact, six of the main enzymes involved in the two main cholesterol synthesis pathways, i.e., Bloch and Kandutsch-Russell pathways [67,68,69,70], are upregulated in hBP-coc compared to hBP-solo. As illustrated in Appendix A [71,72,73,74], Squalene synthase (FDFT1) is involved in early steps of cholesterol synthesis mediating the formation of squalene from farnesyl pyrophosphate. MSMO1, NSDHL and HSD17B7 work in concert to form zymosterol and Δ8-cholestenol in Bloch and Kandutsch-Russell pathways, respectively. EBP mediates the production of Δ7,24-cholestadienol from zymosterol in the Bloch pathway, and lathosterol from Δ8-cholestenol in Kandutsch-Russell pathway. DHCR7 produces desmosterol in Bloch pathway, and is the last enzyme of the Kandutsch-Russell pathway leading to the formation of cholesterol. Surprisingly, we did not measure any variation in the amounts of free cholesterol and cholesteryl in hBP-coc compared with hBP-solo (Figure 6). This observation can be explained by (i) a possible high turnover of cholesterol or a regulation of its synthesis by the sterol responsive element binding protein (SREBP) [75,76,77] as suggested by a GO study and an improvement of this pathway in hBP-coc (Figure 5A), or (ii) its release from BPs. Related to this last point and as we previously described [78], BPs can release free cholesterol to apolipoproteins A–I or E by reverse cholesterol transfer through the transporter ATP-binding cassette family A member 1 (ABCA1, [79]). However, the protein expression of ABCA1 does not seem to vary between hBP-solo and hBP-coc conditions by Western blot (data not shown), and the lysis conditions of the samples used unfortunately do not allow access to data from such transporters with several transmembrane domains—12 for ABCA1 [80]. Scavenger Receptor Class B member 1 or SR-B1 (SCARB1) is a multifunctional receptor capable of importing cholesterol from apolipoproteins or LDL, but also to efflux free cholesterol and cholesteryl esters to HDL or HDL-like lipoparticles [81,82,83]. Since we confirmed that SR-B1 protein expression is increased in hBP-coc and that this receptor is functional according to the improved uptake of free cholesterol from ac-LDL (Figure 6), we suggest that its capacity to release free cholesterol to lipoparticles present in culture media could also be functional. It is also noteworthy that the protein expression of LDLR is increased in hBP-coc compared to hBP-solo. This highlights that hBP are also able to take up free cholesterol through this receptor. Its role in the uptake of ac-LDL remains minimal; however, the fact that these lipoparticles are negatively charged prevents them from interacting with LDLR [84]. In addition, we also pointed-out an enrichment of TIMP3 in hBP-coc condition, which is secreted by BPs to inhibit MMP2/MMP9 and is linked to a maintenance of BBB integrity [85,86]

Altogether, these results suggest that once ECs carry the BBB phenotype, hBPs ‘mature’ or adapt their metabolism to support ECs in maintaining the BBB phenotype by (i) enhancing their energy metabolism and/or (ii) synthesizing cholesterols and sterols for direct use or release. This metabolic gain in hBPs would be of importance to support the NVU cells defects in pathological disorders, such as in Alzheimer’s disease where cholesterol metabolism within the NVU is highly disturbed [87]. However, the fate of thereleased sterols and their mode of delivery to ECs remains unknown but could be helpful to regulate or maintain EC homeostasis and the BBB main features. Moreover, the fact that certain proteins such as Syntenin-1 (SDCBP) and CD63—known to take part in the biogenesis of small extracellular vesicles (EVs) referred to as ‘exosomes’ [88,89,90,91,92]—are overexpressed by hBP-coc is in agreement with a potential cell-cell communication by EVs able to bring protein and lipid regulators to ECs as described between pericytes and ECs at the blood-spinal cord barrier level [93]. We also noticed that short-circuiting the exosome release machinery in BPs endangered the BBB main features at the EC level in vitro (unpublished data). Although these points need to be explored further, this EV-mediated communication could also partly explain (i) the cholesterol requirement of BPs since the biogenesis mechanisms of EVs are intimately linked with the dynamics of membrane lipids and phospholipids [94], and (ii) why the cholesterol level of cocultured BPs does not vary despite the significant increase in protein expression of key enzymes of its de novo synthesis.

## 5. Conclusions

In conclusion, our work in a non-contact human BBB model highlighted in vitro the change in the proteomic profile of hBPs induced by ECs when they acquire the BBB phenotype, shifting from a ‘naïve’ or ‘immature’ state related to cell proliferation and contractility, to a ‘mature’ or ‘differentiated’ state with a deep metabolic switch in favor of energy and lipid metabolism (Figure 7). Thus, just as ECs benefit from BPs for their differentiation into BBB ECs, hBPs also benefit from ECs for their own cell differentiation into what we can refer to as ‘BBB BPs’. As potential outcomes of this study, it would be interesting to compare these data with a contact human BBB model to consider all the aspects of BPs-BBB ECs interactions. These bidirectional exchanges and this bidirectional maturation are essential to ensure the maintenance and homeostasis of the BBB in an in vitro coculture model and, from a more global point of view, of the NVU in vivo.

## Figures and Tables

**Figure 1 cells-12-01010-f001:**
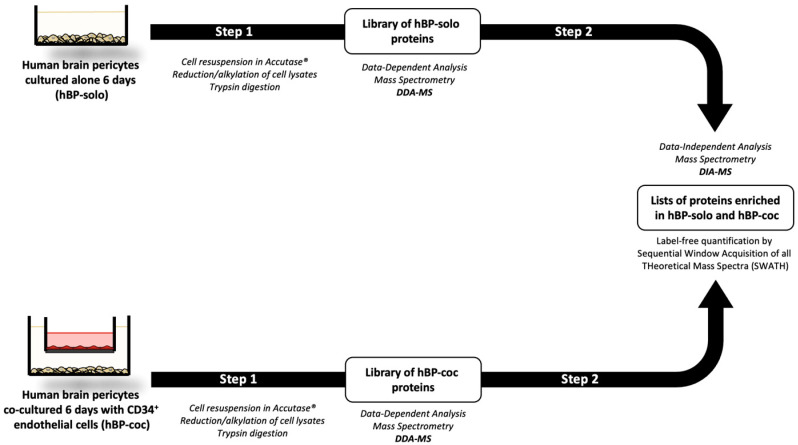
Graphical experimental design of the study. Human brain pericytes (hBP) are processed for mass spectrometry analysis after 6 days alone (hBP-solo) or in coculture with CD34^+^ endothelial cells (hBP-coc). 6 days correspond to the time necessary to promote the BBB main features on in the coculture model used, and therefore to optimize bidirectional cell-cell communications along the BBB establishment. Step 1 refers to the setup of protein libraries in both conditions (DDA-MS), Step 2 corresponds to the generation of label-free SWATH quantification of proteins enriched in hBP-solo or in hBP-coc conditions (DIA-MS analysis).

**Figure 2 cells-12-01010-f002:**
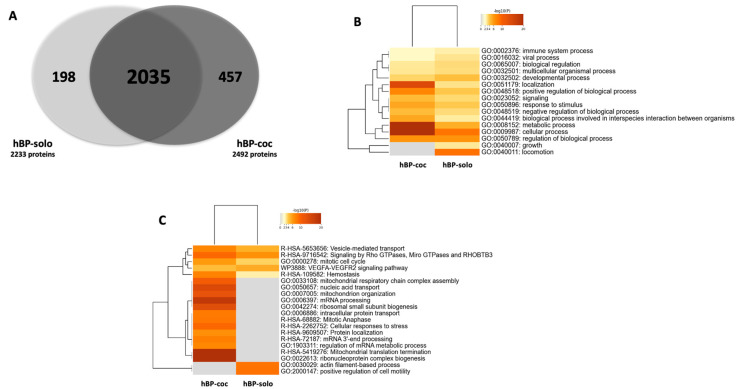
Libraries and functions of proteins identified as enriched in hBP-solo and hBP-coc conditions. (**A**) Venn diagram of the identified proteins in hBP-solo and hBP-coc conditions by DDA-MS analysis (2233 and 2492 respectively, N = 2, *n* = 10). Only proteins identified based on at least 2 peptides were included in this study. (**B**) Top-level gene ontology (GO) biological processes for hBP-solo and hBP-coc enriched proteins. (**C**) Dendogram of statistically enriched ontology clusters for hBP-solo and hBP-coc enriched proteins. −log10(*p*-values) represents the enrichment of protein candidates in each cluster.

**Figure 3 cells-12-01010-f003:**
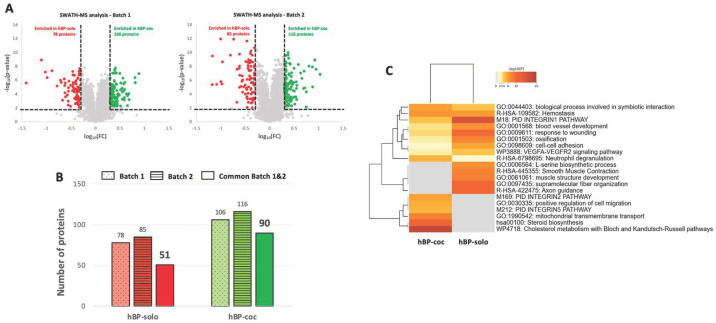
SWATH quantification of proteins enriched in hBP-solo and hBP-coc. (**A**) Volcano plots from the two independent batches used for the study. Inclusion criteria to point out significant candidates: log10(Fold Change) > 0.3 for hBP-coc enriched proteins (FC > 2); log10(FC) < 0.3 for hBP-solo enriched proteins (FC < 2); −log10(*p*-value) < 0.01 for the selected proteins in both conditions (*p*-value < 0.01). (**B**) Number of significant enriched protein in hBP-solo and hBP-coc from the 2 independent batches and common between the two batches. The reduced number of proteins in this common pool corresponds to the mean of SWATH quantification between Batch 1 and Batch 2 with FC > 2. (**C**) Dendogram of statistically enriched ontology clusters for significant hBP-solo and hBP-coc enriched proteins. −log10(*p*-values) represents the enrichment of protein candidates in each cluster.

**Figure 4 cells-12-01010-f004:**
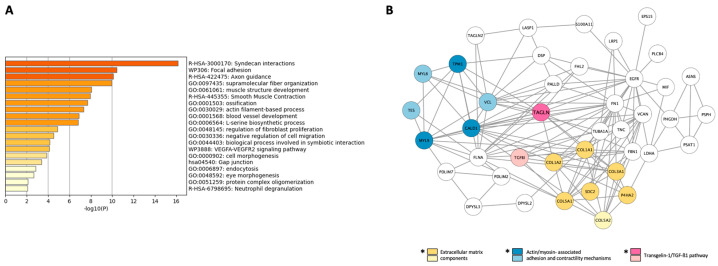
Transgelin 1 (TAGLN)-mediated processes are improved in hBP-solo. (**A**) Top-level gene ontology (GO) analysis and (**B**) Interaction map of the 51 proteins quantitatively enriched in hBP-solo. *: Top-20 proteins.

**Figure 5 cells-12-01010-f005:**
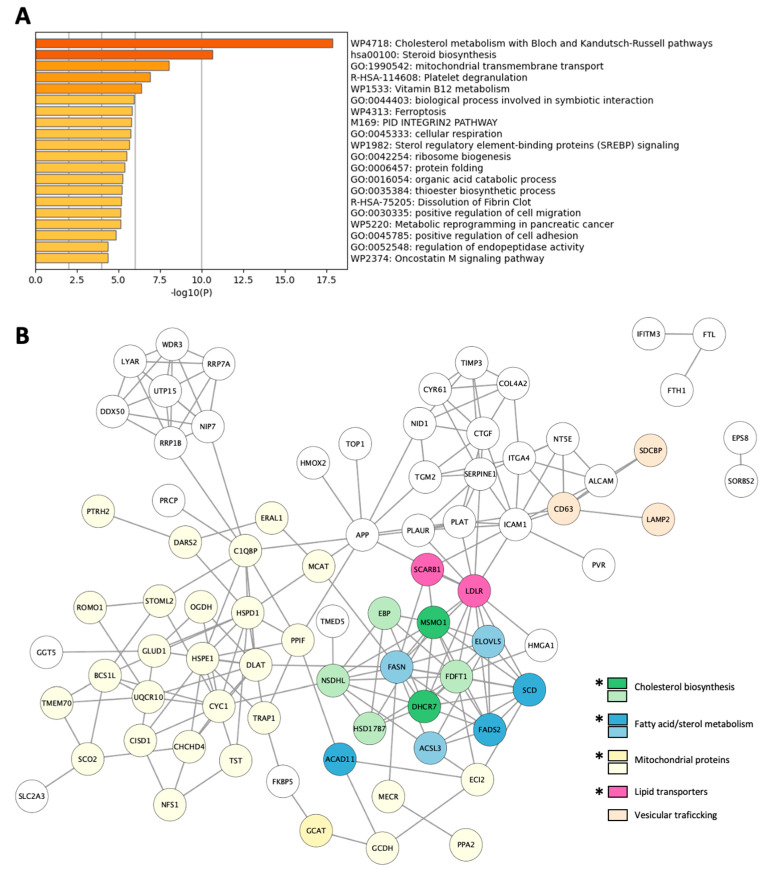
Improvement of cholesterol, fatty acids and sterols metabolism, and mitochondrial functions in hBP-coc. (**A**) Top-level gene ontology analysis and (**B**) Interaction map of the 90 proteins quantified in hBP-coc. *: Top-20 proteins.

**Figure 6 cells-12-01010-f006:**
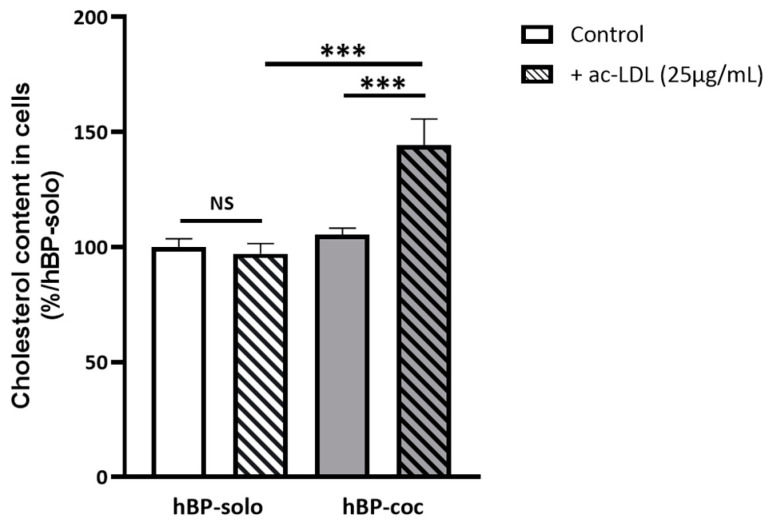
Cholesterol content in hBP-coc is increased after acetylated LDL (ac-LDL) treatment only. After 6 days of culture, hBP-solo and hBP-coc were treated 3 h with or without 25 µg/mL of ac-LDL and their cholesterol content were measured and normalized by µg of proteins. Each bar represents the mean ± SEM (*n* = 6–9). Statistical analysis: One-way ANOVA with Bonferroni’s multiple comparison test, NS: non-significant, ***: *p* < 0.001.

**Figure 7 cells-12-01010-f007:**
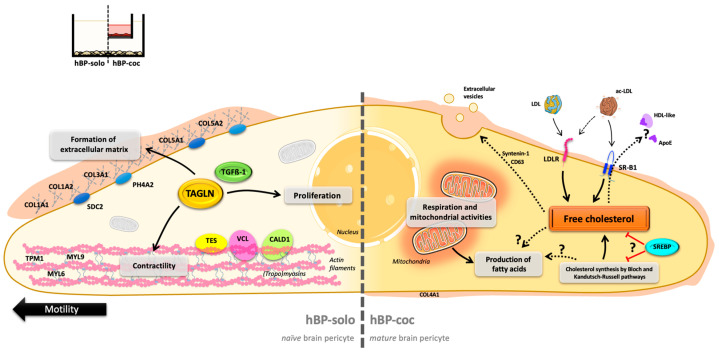
Prospective conclusions—Brain pericyte proteome is switching from naïve to mature owing to the influence of BBB ECs. Briefly, hBP-solo exhibit immature features linked to cell proliferation, motility (black arrow), contractility, and the formation and secretion of extracellular matrix components such as collagen fibers and proteoglycans (SDC2 and PH4A2). Under the influence of ECs, hBP-coc matured their metabolical functions, particularly mitochondrial activities and sterols and cholesterol homeostasis. Acronyms: ApoE: Apolipoprotein E; CALD1: Caldesmon; COL1-3-4-5A1: Collagen alpha-1 (I, III, IV, V) chain; COL1-5A2: Collagen alpha-2(I,V) chain; HDL: high density lipoproteins; LDL: low-density lipoproteins; ac-LDL: acetylated LDL; LDLR: LDL receptor; MYL6-9: Myosin regulatory light polypeptide 6-9; PH4A2: Prolyl 4-hydroxylase subunit alpha-2; SDC2: Syndecan-2; SR-B1: Scavenger Receptor class B member 1; SREBP: Sterol Responsive Element Binding Protein; TAGLN: Transgelin; TES: Testin; TGFß-1: Transforming Growth Factor ß-1; TPM1: Tropomyosin alpha-1 chain; VCL: Vinculin.

**Table 1 cells-12-01010-t001:** Top-20 of the 51 proteins enriched in solo-cultured hBP. The complete list is available in Appendix A.

Accession Number	Gene Name	Detailed Name	Mean ^a^	S.D.
P02452	COL1A1	Collagen alpha-1(I) chain	13.8	1.62
P08123	COL1A2	Collagen alpha-2(I) chain	9.92	0.910
P02461	COL3A1	Collagen alpha-1(III) chain	9.89	0.257
Q01995	TAGLN	Transgelin	7.03	0.979
P24844	MYL9	Myosin regulatory light polypeptide 9	4.67	0.737
Q9Y617	PSAT1	Phosphoserine aminotransferase	4.19	0.098
P15924	DSP	Desmoplakin	3.62	0.07
P09493	TPM1	Tropomyosin alpha-1 chain	3.57	1.12
P42566	EPS15	Epidermal growth factor receptor substrate 15	3.51	0.814
Q05682	CALD1	Caldesmon	3.43	0.973
P24821	TNC	Tenascin	3.42	1.09
Q5EB52	MEST	Mesoderm-specific transcript homolog protein	3.29	0.613
P20908	COL5A1	Collagen alpha-1(V) chain	3.27	0.398
Q8WX93	PALLD	Palladin	3.22	0.481
P78330	PSPH	Phosphoserine phosphatase	3.21	0.518
Q71U36	TUBA1A	Tubulin alpha-1A chain	3.18	0.393
Q96IZ0	PAWR	PRKC apoptosis WT1 regulator protein	3.17	0.641
O15460	P4HA2	Prolyl 4-hydroxylase subunit alpha-2	3.16	0.345
P34741	SDC2	Syndecan-2	3.13	1.343
P08243	ASNS	Asparagine synthetase [glutamine-hydrolyzing]	2.99	0.321

^a^ Mean from the two independent SWATH analyses.

**Table 2 cells-12-01010-t002:** Top 20 of the 90 proteins enriched in cocultured hBP. The complete list is available in Appendix A.

Accession Number	Gene Name	Detailed Protein Name	Mean ^a^	S.D.
P21980	TGM2	Protein-glutamine gamma-glutamyltransferase 2	7.57	0.21
Q13201	MMRN1	Multimerin-1	7.35	1.30
P05362	ICAM1	Intercellular adhesion molecule 1	6.97	0.84
P08572	COL4A2	Collagen alpha-2(IV) chain	4.34	1.00
P35625	TIMP3	Metalloproteinase inhibitor 3	4.07	0.29
P00750	PLAT	Tissue-type plasminogen activator	4.03	0.64
P52566	ARHGDIB	Rho GDP-dissociation inhibitor 2	4.03	0.77
Q15800	MSMO1	Methylsterol monooxygenase 1	4.01	1.08
O00622	CYR61	CCN family member 1	3.92	0.11
O94919	ENDOD1	Endonuclease domain-containing 1 protein	3.92	0.94
O00767	SCD	Stearoyl-CoA desaturase	3.91	0.57
P01130	LDLR	Low-density lipoprotein receptor	3.47	0.40
Q13451	FKBP5	Peptidyl-prolyl cis-trans isomerase FKBP5	3.44	0.22
Q8WTV0	SCARB1	Scavenger receptor class B member 1	3.37	1.632
P36269	GGT5	Glutathione hydrolase 5 proenzyme	3.28	0.739
Q9UBM7	DHCR7	7-dehydrocholesterol reductase	3.15	0.001
O95864	FADS2	Acyl-CoA 6-desaturase	3.13	0.514
O75600	GCAT	2-amino-3-ketobutyrate coenzyme A ligase, mitochondrial	3.05	0.656
Q01628	IFITM3	Interferon-induced transmembrane protein 3	3.01	0.099
Q709F0	ACAD11	Acyl-CoA dehydrogenase family member 11	2.92	0.093

^a^ Mean from the two independent SWATH analyses.

## Data Availability

All the datasets used and/or analyzed during the current study are available as Appendix A.

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
