# Peer review of "Highlighting In Vitro the Role of Brain-like Endothelial Cells on the Maturation and Metabolism of Brain Pericytes by SWATH Proteomics"

_cells, 2023, doi:10.3390/cells12071010_

Round 1

Reviewer 1 Report

The manuscript by Menaceur et al., describes a proteomic comparison of human pericytes either solo-cultured or co-cultured with human endothelial cells. 

The study is novel, clearly-written and well-presented, and the findings will be of general interest and specific interest to those in the field.

Major comments:

1. The robustness of the study could be improved with inclusion of data that fully characterise the pericyte population used in this study (sorry if I missed this).

Although the pericytes are pretty-well characterised in Ref [31], details such as passage number and cell number used in this study are missing from the methodology.

Have they been labelled for any pericyte markers in the lab - could these images be shown in a supplementary figure? What passage number has been used and have the cells been assessed for expression of fibroblast markers?

Since these cells express a-SMA [ref 31], how can you be sure that they are pericytes since pericytes represent a heterogenous collection of cells and only a subset of contractile pericytes on pre-capillary arterioles express a-SMA (perhaps this could be expanded in the discussion) 

Minor comments: 

1. LRP-1 and TIMP3 were in the top 20 proteins enriched in the co-culture that play an important role in pericyte-mediated uptake and phagocytosis of beta-amyloid, and regulation of BBB permeability, respectively. Given the prominent role of TIMP-3 and LRP-1, could this findings be highlighted and  expanded in the discussion

1. Highlight that a non-contact model is being used in the abstract

Author Response

The manuscript by Menaceur et al., describes a proteomic comparison of human pericytes either solo-cultured or co-cultured with human endothelial cells. 

The study is novel, clearly-written and well-presented, and the findings will be of general interest and specific interest to those in the field.

The authors would like to thank the Reviewer 1 for his pertinent report and his nice feeling regarding the study and the quality of the manuscript.

Major comments:

1. The robustness of the study could be improved with inclusion of data that fully characterise the pericyte population used in this study (sorry if I missed this).

The authors thank the reviewer for this remark. The main goal of this study is to analyze human brain pericytes through a proteomics approach and according to the DDA data, we are pleased to bring a deeper characterization of this pericyte population. Itself, the proposed study is thus a way to characterize these cells. The complete lists of identified proteins in hBP-solo and hBP-coc are detailed in Table S1 and Table S2.

2. Although the pericytes are pretty-well characterised in Ref [31], details such as passage number and cell number used in this study are missing from the methodology.

Human brain pericyte cell line was used between passage 15 and 25 according to the recommendations of Pr. Kanda who isolated and immortalized the cells [31]. Moreover, 6.25x105 hBP were seeded in 100mm Petri dishes for both solocultured and co-cultured conditions. We added this information in Line 89.

3. Have they been labelled for any pericyte markers in the lab - could these images be shown in a supplementary figure? What passage number has been used and have the cells been assessed for expression of fibroblast markers?

We previously published a study using this hBP cell line with IF for so-called markers such as Desmin, PDGFRß and α-SMA (Deligne et al., 2020, Fluids and Barrriers of the CNS). Please find below the corresponding part a) of Figure 1:

We also added in the text a sentence linked to this previous result.

Line 87-88: “The hBP cell line was previously characterized for so-called pericyte markers such as Desmin, PDGFRß and α-SMA [32].”

4. Since these cells express a-SMA [ref 31], how can you be sure that they are pericytes since pericytes represent a heterogenous collection of cells and only a subset of contractile pericytes on pre-capillary arterioles express a-SMA (perhaps this could be expanded in the discussion) 

The authors thank the reviewer for this suggestion. Based on the reference [31], published in 2011, α-SMA was considered as a “brain pericyte marker”. However, in the last 12 years, knowledge regarding brain pericyte subpopulation arose to highlight how heterogenous the pericytes are all along the brain (micro)vasculature and established at least three pericyte subpopulations (reference [53] in the manuscript). According to these fresh studies, α-SMA can’t be considered as a so-called brain pericyte marker since this protein is also expressed by smooth muscle cells. However, we also identified PDGFR-ß, Desmin and pericyte-specific MYLs such as MYL6 and MYL9 in our study, suggesting that the cells we used are brain pericyte-like cells.

Minor comments: 

1. LRP-1 and TIMP3 were in the top 20 proteins enriched in the co-culture that play an important role in pericyte-mediated uptake and phagocytosis of beta-amyloid, and regulation of BBB permeability, respectively. Given the prominent role of TIMP-3 and LRP-1, could this findings be highlighted and expanded in the discussion

We clarified the role of TIMP3 in hBP-coc condition as proposed by the Reviewer 1 (Lines 494-496). However, we assume not to discuss about the role of LRP1 since (i) that receptor is enriched in hBP-solo and not in hBP-coc and (ii) do not appear in any top-20 lists.

2. Highlight that a non-contact model is being used in the abstract

We added this information in the abstract (Line 20).

Reviewer 2 Report

This manuscript reports on experiments designed to reveal the interactions of brain endothelial cells (BEC) with brain pericytes both of which are key cellular components of the neurovascular unit (NVU). Rather than examining the influence of brain endothelial cells on brain pericytes (PC), the authors focus on the other way around and examine the influence of BECs on BPs. The approach involves in vitro studies of cultured BPs alone or BPs co-cultured with BECs in a Transwell plate. BPs under each condition (solo culture or co-culture) undergo protein extraction and analysis by LC-MS/MS and subsequent quantitation and comparisons. The results are further analyzed using GO analysis and related protein identification and expression analysis to conclude that ECs cause a shift in BPs from a more primitive or naïve state to a more differentiated state with significant mitochondrial activity and cholesterol metabolism activity.

The purpose of the study is to examine cell-cell communication responses between brain endothelial cells and brain pericytes with a focus on BPs of the NVU and the influence of brain ECs. The study is conducted in vitro with model cells cultured under previously developed conditions. The design is straightforward and described with a figure. The primary analysis is the use of proteomic approaches. Cultured cells are subject to protein extraction and then analysis by LC-MS/MS techniques for identification and analysis. The methods and instrumentation for this are described in sufficient and good detail for another laboratory to pursue and/or replicate. The results are presented in a tables, gene ontology lists and interaction maps of the top proteins. The authors find that co-culture of BECs with BPs induces the BPs to shift from a more undifferentiated state to a more active state based on protein expression analysis and ontology analysis.

There are a few minor concerns or suggestions for strengthening the report.

  1. This approach seems to favor the most abundant proteins and underestimate the importance of lesser abundant proteins. If so, this feature should be mentioned as a potential factor affecting the interpretations.
  2. In the Discussion, section 4.1. A significant factor may be the effect of blood flow. There are agonists for stimulating flow receptors. This might be considered or mentioned in this section.
  3. Discussion, lines 506-512. This section mentions an altered phenotype and the authors may wish to address the metabolic substrates or altered energy demands of cells in the NVU.
  4. Discussion, lines 515-517. This sentence (We also seen….) appears to be missing something and should be corrected.
  5. Discussion, lines 517-523. It is intriguing that fatty acid and cholesterol metabolism are noted. Because of the importance of these pathways in neurogenerative conditions, it I recommended that some attention is placed on these conditions such as the potential role of APOEs, cholesterol metabolism and Alzheimer’s Disease or other diseases.

Author Response

The authors thank the Reviewer 2 for his pertinent report and his nice feeling regarding the study and the quality of the manuscript.

There are a few minor concerns or suggestions for strengthening the report.

1. This approach seems to favor the most abundant proteins and underestimate the importance of lesser abundant proteins. If so, this feature should be mentioned as a potential factor affecting the interpretations.

The authors thank the Reviewer 2 for this remark. We assume the strict conditions applied to point-out so-called quantitatively significant protein candidates (p-value < 0.01 and FC > 2 or FC < 2). These conditions are kind of classical to improve the statistical strength of the study and well-adapted to SWATH analysis. We obviously underestimate some proteins with biological relevance; however, they can’t be taken into account as significant. That can explain why we didn’t focus on proteins under these strict criteria.

2. In the Discussion, section 4.1. A significant factor may be the effect of blood flow. There are agonists for stimulating flow receptors. This might be considered or mentioned in this section.

We considered this remark and added a short part of sentence in section 4.1.

Lines 410-412: “However, this point remains to be further investigated to better define the mode and mechanisms of this regulation by ECs, and how cerebral blood flow could also impact hBP physiology.”

3. Discussion, lines 506-512. This section mentions an altered phenotype and the authors may wish to address the metabolic substrates or altered energy demands of cells in the NVU.

The authors assume that the modified phenotype between solocultured and co-cultured hBP with ECs is not actually an alteration of metabolism but a way for hBP to mature. However, we stated that these metabolic changes could be of interest to support the NVU cells are altered to pathological disorders. We added a sentence in the corresponding paragraph.

Lines 514-516:” This metabolic gain in hBPs would be of importance to support the NVU cells defects in pathological disorders, such as in Alzheimer’s disease where cholesterol metabolism within the NVU is highly disturbed [88]. However, the fate of the …”

4. Discussion, lines 515-517. This sentence (We also seen....) appears to be missing something and should be corrected.

We apology for that sentence, we modified it since some words were mistaken (‘short-cutting’ used instead of ‘short-circuiting’).

Lines 523-525: “We also noticed that short-circuiting the exosome release machinery in BPs endangered the BBB main features at the EC level in vitro (unpublished data).”

5. Discussion, lines 517-523. It is intriguing that fatty acid and cholesterol metabolism are noted. Because of the importance of these pathways in neurogenerative conditions, it I recommended that some attention is placed on these conditions such as the potential role of APOEs, cholesterol metabolism and Alzheimer’s Disease or other diseases.

We agree with this argument, and some of the authors already published studies in that way. We mentioned that idea in a small sentence earlier in the discussion and added a reference linked to ApoE4 and AD (reference 88 Montagne A. et al., 2020 Nature).

Lines 514-516:” This metabolic gain in hBPs would be of importance to support the NVU cells defects in pathological disorders, such as in Alzheimer’s disease where cholesterol metabolism within the NVU is highly disturbed [88]. However, the fate of the …”

Reviewer 3 Report

I really enjoyed reading this article, which gives biological significance to the proteomic results obtained on BPs solocultured and co-cultured with ECs in relation to the human BBB modelling. The paper is worthy of publication, is pleasant to read and of great interest.

I have, however, some comments and suggestions that, in my opinion, can improve the quality of the paper.

1) I think it is important, when a proteomic study so rich in content has been performed, to make available to scientific comunity the MSMS data, submitting them on a free repository. I invite the authors to do it. It is procedure that does not require much time and above all avoids attaching many supplementary files  

2) page 6, from line 230: Why the identification was performed by DDA and the label-free in DIA mode? Why not identify and quantify proteins directly in DIA by SWATH-MS analysis?  Can you explain better the advantages of having used more steps and two identification methods?

 3) Peptides (and thus proteins) identified in DDA are the same identified and quantified in DIA? It is not apparent from how the results are presented. Improve this aspect. 

also page 7, from line 282: did you quantify proteins previously identified in DDA mode? Or did you quantify different proteins identified by DIA mode? Clarify if there is a correspondence between identifications in Table S1 and S2. Add the information in the Tables. Discuss in the text

Author Response

I really enjoyed reading this article, which gives biological significance to the proteomic results obtained on BPs solocultured and co-cultured with ECs in relation to the human BBB modelling. The paper is worthy of publication, is pleasant to read and of great interest.

The authors thank the Reviewer 3 for his pertinent report and his nice feeling regarding the study and the quality of the manuscript.

I have, however, some comments and suggestions that, in my opinion, can improve the quality of the paper.

1) I think it is important, when a proteomic study so rich in content has been performed, to make available to scientific comunity the MSMS data, submitting them on a free repository. I invite the authors to do it. It is procedure that does not require much time and above all avoids attaching many supplementary files  

The authors are aware of this consideration. However, it is noteworthy that all the raw data of this study are available in supplementary data and that the actual condition to deposit MSMS data on a free repository are time consuming. In the current state, we assume to propose the data as Supplementary Materials, but we actively consider submitting our next proteomics data in a repository.

2) page 6, from line 230: Why the identification was performed by DDA and the label-free in DIA mode? Why not identify and quantify proteins directly in DIA by SWATH-MS analysis?  Can you explain better the advantages of having used more steps and two identification methods?

 3) Peptides (and thus proteins) identified in DDA are the same identified and quantified in DIA? It is not apparent from how the results are presented. Improve this aspect. also page 7, from line 282: did you quantify proteins previously identified in DDA mode? Or did you quantify different proteins identified by DIA mode? Clarify if there is a correspondence between identifications in Table S1 and S2. Add the information in the Tables. Discuss in the text.

The authors thank the Reviewer 3 for remarks 2 and 3. Using DDA mode, we first construct the peptide library which includes the exact m/z values (both precursor ion and product ion), retention time, and protein name identified by exact m/z values of precursor ion and product ion. Then by analyzing the sample with DIA mode (SWATH acquisition mode), we get the information of the exact m/z value of "product ion" for each peptide, its retention time, and peak area. Regarding the m/z value of precursor ion, we can only know the mass "range", e.g., m/z 200-500 in the SWATH acquisition mode. Therefore, we are able to identify the peptide in SWATH data using library data created by DDA, which means that the peptide detected in SWATH acquisition mode cannot be identified by using only the DIA data. The advantages of this two-step procedure are that (i) we can detect the peptides with low abundance in the sample and (ii) we can quantify the peptide using several product ions produced from one peptide. The comprehensiveness and the quantification force are therefore increased.

Round 2

Reviewer 1 Report

Thank you, I am satisfied that the authors have adequately addressed my initial comments.

Reviewer 3 Report

I think that the paper can be accelererà for the pubblication